# Seasonal transcriptomic shifts reveal metabolic flexibility of chemosynthetic symbionts in an upwelling region

Isidora Morel-Letelier,[1] Benedict Yuen,[1] Luis H. Orellana,[2] A. Carlotta Kück,[1] Yolanda E. Camacho-García,[3,4,5] Minor Lara,[6] Matthieu Leray,[7] Laetitia G. E. Wilkins[1]

**ABSTRACT** Upwelling in the Tropical Eastern Pacific profoundly affects marine coastal ecosystems by driving drastic seasonal changes in water temperature, oxygen levels, and nutrient availability. These conditions serve as a natural experiment that provides a unique opportunity to study how marine animals and their associated microorganisms respond in the face of environmental change. Lucinid bivalves host chemosynthetic *Candidatus* Thiodiazotropha symbionts equipped with diverse metabolic pathways for sulfur, carbon, and nitrogen use. However, how these symbionts employ their metabolic toolkit in a changing environment remains poorly understood. To address this question, we conducted metagenomic and metatranscriptomic analyses of *Ctena* cf. *galapagana* symbionts before and during a Papagayo upwelling event in Santa Elena Bay, Costa Rica. The *C.* cf. *galapagana* were co-colonized mainly by two *Ca.* Thiodiazotropha symbiont clades regardless of the sampling season. We observed a concerted shift in the transcriptomic profiles of both symbiont clades before and during upwelling, suggesting changes in energy source use. Dissimilatory methanol oxidation genes were upregulated before upwelling, while sulfide oxidation genes were upregulated during upwelling. These physiological changes were potentially driven by upwelling-induced changes in sediment biogeochemistry and resource availability. Our findings highlight the adaptability of the lucinid symbiosis and the crucial role of symbiont metabolic flexibility in their resilience to environmental challenges.

**IMPORTANCE** The oceans are undergoing rapid change, and marine animals together with their associated microorganisms must adjust to these changes. While microbes are known to play a critical role in animal health, we are only beginning to understand how symbiotic relationships help animals cope with environmental variability. Annual upwelling events cause drastic and abrupt increases in nutrient availability and productivity, while temperature and oxygen decrease. In this study, we investigated how bacterial symbionts of the lucinid bivalve *Ctena* cf. *galapagana* respond to upwelling in the Tropical Eastern Pacific. The symbionts, from the genus *Candidatus* Thiodiazotropha, are chemosynthetic (i.e., they use inorganic chemicals for energy and fix carbon) and provide nutrition to their host. Our results show that these symbionts adjust their use of different energy sources in response to environmental changes that affect resource availability. This metabolic flexibility underscores the resilience of animal-microbe relationships in coping with environmental change.

**KEYWORDS** symbiosis, marine, lucinid, upwelling, environmental change, host-microbe interactions, metatranscriptomics, metagenomics, microbial metabolism

Animals form intimate associations with microorganisms—that is, bacteria, archaea, viruses, fungi, and microeukaryotes—that play a pivotal role in their evolution, ecology, and development (1). A host's metabolic repertoire can shift or expand through

Address correspondence to Isidora Morel-Letelier, imorel@mpi-bremen.de, or Laetitia G. E. Wilkins, megaptera.helvetiae@gmail.com.

The authors declare no conflict of interest.

See the funding table on p. 15.

the acquisition, loss, or replacement of microorganisms, as well as through changes in the expression of microbial genes (2–4). Together, these changes in the associated microbial community can enhance a host's tolerance of environmental changes through a process known as "microbiome-mediated acclimatization" (5–7). This mechanism could potentially play a vital role in enabling many organisms to adapt to the unprecedented environmental changes currently affecting the planet (8). Conceptual frameworks on how microorganisms may extend hosts' phenotypes and evolutionary potential have been developed (9). However, empirical studies investigating the responses of host-associated microorganisms to environmental changes and their effects on the host remain limited (10, 11). Further progress has also been challenged by the diversity and poor functional characterization of many animal-associated microbial assemblages.

Animals relying on specialized microbial partnerships provide tractable systems to study microbiome-mediated acclimatization (12). Bivalves of the family Lucinidae (lucinids) form obligate symbiotic relationships with chemosynthetic Gammaproteobacteria, most commonly from the genus *Candidatus* Thiodiazotropha (13–15), the primary symbiont lineage. Juvenile lucinids acquire these symbionts horizontally from free-living populations (16) and host them in specialized gill cells (17) where they oxidize reduced sulfur compounds to fuel $CO_2$ fixation into organic carbon (18), which is then transferred to the host as a primary nutrient source (19, 20). Lucinids supply their symbionts with oxygen from the water column and reduced sulfur compounds mined from deeper sediments via their extensible foot (21–24). Lucinids have diversified into a wide range of marine environments worldwide, including shallow seagrass meadows, rocky intertidal zones, and the deep sea. Recent studies have highlighted the functional diversity of *Ca*. Thiodiazotropha symbionts across these habitats, revealing a broad repertoire of metabolic pathways. These include the genomic potential to oxidize sulfur compounds via multiple pathways (25), as well as the ability to oxidize methanol (14) and/or hydrogen (26)—both as energy sources and, in the case of methanol, also as an additional carbon source (14, 27, 28). Adaptation to variable oxygen conditions is also reflected by the potential of certain lineages to use oxygen and various nitrogen compounds as terminal electron acceptors (14). Furthermore, the capacity for nitrogen fixation has been identified in symbionts from oligotrophic environments (26, 29). However, our understanding of how these diverse metabolic pathways are regulated under changing environmental conditions, particularly those affecting the availability of essential resources, remains limited.

Seasonal upwelling in the Tropical Eastern Pacific (TEP) plays a critical role in shaping the environmental conditions of marine ecosystems along the Pacific coasts of Central America (30). Each year during the dry season, strong wind jets originating from the Caribbean cross the Cordillera mountains near Lake Nicaragua. These winds displace warm, nutrient-poor surface waters offshore, allowing cold, nutrient-rich deep waters to rise to the surface (31, 32). This process significantly cools surface temperatures and increases concentrations of nutrients, such as phosphate, ammonia, and nitrate, sometimes by as much as 15-fold, triggering intense phytoplankton blooms and increasing particulate organic carbon in the water column (33). As this organic matter is remineralized, microbial aerobic respiration in both the water column and underlying sediments intensifies, rapidly consuming available oxygen. The resulting micro-oxic or anoxic conditions promote anaerobic microbial processes, including denitrification, anammox, sulfate reduction, and methanogenesis (reviewed in reference 34). When winds weaken during the wet season, surface waters return to a warmer, nutrient-poor, and chlorophyll-depleted state. These seasonal changes profoundly affect coastal marine ecosystems in the TEP, influencing benthic macroalgal abundance (35), fish community composition (36), and the physiology and density of coral symbionts (37). Lucinids that inhabit these environments are also subject to the contrasting environmental conditions brought about by the TEP upwelling system, and thus this system provides a natural experiment to explore how the metabolic toolkit of the lucinid symbionts is deployed in response to environmental change.

To investigate the response of lucinid symbionts to seasonal upwelling, we sequenced the metagenomes and metatranscriptomes of *Ctena* cf. *galapagana* individuals sampled at the rocky intertidal shore of Santa Elena Bay, Costa Rica, before (November 2022) and during (March 2023) the Papagayo upwelling. Metagenomic analysis revealed that primary symbiont abundance and composition, dominated by two clades of the genus *Candidatus* Thiodiazotropha, remained stable across seasons. The analysis of the metatranscriptomes revealed differential expression of pathways involved in energy acquisition. These physiological changes are likely driven by upwelling-induced alterations in inorganic matter deposition, sediment biogeochemistry, and the availability of energy-rich compounds. Our findings provide novel insights into the crucial role of symbiont metabolic flexibility in response to environmental change.

## RESULTS

### Seasonal upwelling had a strong effect on primary productivity in Santa Elena Bay

We sampled *Ctena* cf. *galapagana* in Santa Elena Bay (Fig. 1A) during a non-upwelling period in November 2022 (nUPW) and during the upwelling season in March 2023 (UPW) (Fig. 1B). Remote sensing measurements of Santa Elena Bay revealed the effects of the upwelling peaked in February at an average daily chlorophyll concentration of $3.72 \pm 0.45$ mg/m$^3$ (mean $\pm$ standard deviation), which was fourfold greater than in November (non-upwelling) and less than two times higher than in March (upwelling) (Fig. 1B; Table S1). The elevated chlorophyll concentrations in February coincided with an increase in nitrate concentrations and a decrease in both water potential temperature and oxygen concentration (Fig. 1B). The non-upwelling chlorophyll levels in November ($0.86 \pm 0.45$ mg/m$^3$) were less than half those observed during upwelling in March ($2.16 \pm 0.9$ mg/m$^3$) (Fig. 1B; Table S1), thereby establishing the contrasting conditions of the experiment. Oxygen concentrations and water temperature were similar at both sampling time points but were more variable throughout the upwelling season.

### Upwelling conditions did not alter primary symbiont abundance or composition

We sequenced the metagenomes of 10 lucinids sampled in the non-upwelling and upwelling seasons, respectively, to explore how changing environmental conditions affect the composition and abundance of the primary symbionts (Table S2). We retrieved nine symbiont metagenome-assembled genomes (MAGs) from nine upwelling metagenomes (Table S3), three of which were high quality (>95% completeness, <5% contamination). Two high-quality MAGs were classified as *Ca*. T. boucheti (average nucleotide identity [ANI] ~99.2% to reference MAGs) and one as *Ca*. T. endolucinida TEP (ANI ~99.8% to reference MAGs; Table S4). None of the three MAGs recovered from the non-upwelling metagenomes were high quality (Table S3).

To further investigate the effects of upwelling on primary symbiont composition, we mapped the metagenomic reads of the lucinids sampled in non-upwelling and upwelling to reference MAGs (Table S3) of *Ca*. T. boucheti, *Ca*. T. endolucinida TEP, and *Ca*. T. larai, the three symbiont species clades previously described in this region (15). Regardless of the sampling season, most of the metagenome libraries comprised roughly equal proportions of reads belonging to *Ca*. T. boucheti and *Ca*. T. endolucinida TEP, while only a few samples contained reads belonging to just one of these two symbiont species alone (Fig. 2A; Table S5). Reads belonging to *Ca*. T. larai were detected only in one library at low abundance compared to the other two clades (Fig. 2A). We used the total normalized sequencing depth of the symbionts' genomes (TNSD) as a proxy for estimating symbiont abundance (38, 39). The median, mean, minimum, and maximum values of the TNSD in the non-upwelling samples were lower compared to upwelling samples, but this difference was not statistically significant ($P \geq 0.05$) (Fig. 2B; Table S5).

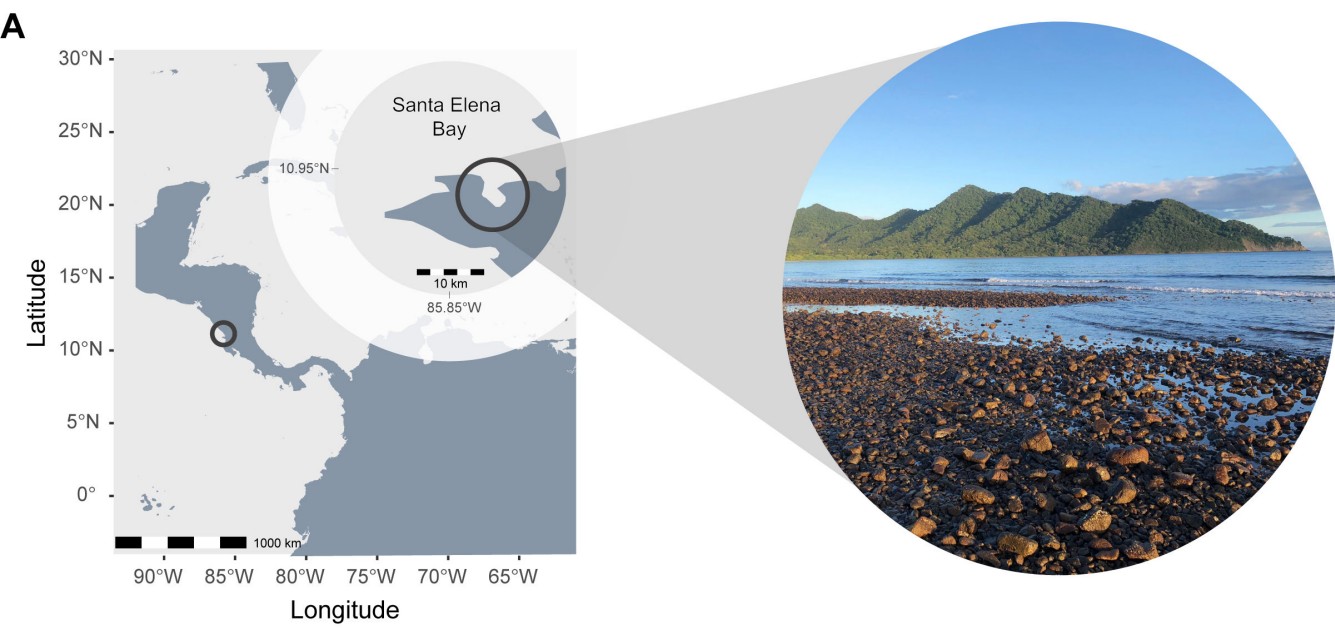

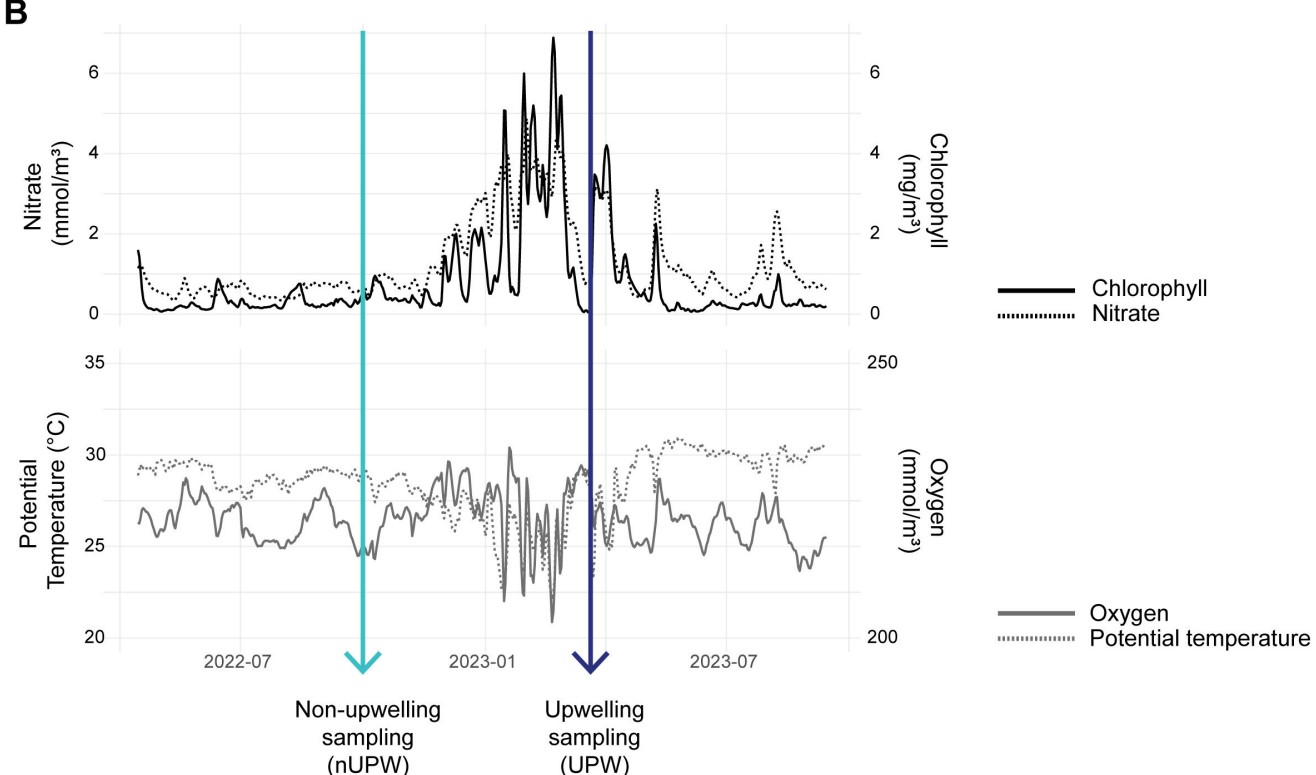

**FIG 1** *Ctena* cf. *galapagana* in Santa Elena Bay, Costa Rica, were sampled in the non-upwelling season in November and just after the peak of the upwelling season in March. (A) Sampling took place in the Tropical Eastern Pacific, Santa Elena Bay, Costa Rica. (© Isidora Morel-Letelier. This image is openly licensed via CC BY 4.0.) (B) Effects of seasonal upwelling in Santa Elena Bay. Remote sensing measurements of the water column in Santa Elena Bay obtained from the Copernicus database. Based on chlorophyll, oxygen, and temperature measurements, the effects of upwelling peaked in February 2023. Non-upwelling nitrate and chlorophyll concentrations in November (turquoise arrow) were considerably lower than during upwelling in March (blue arrow).

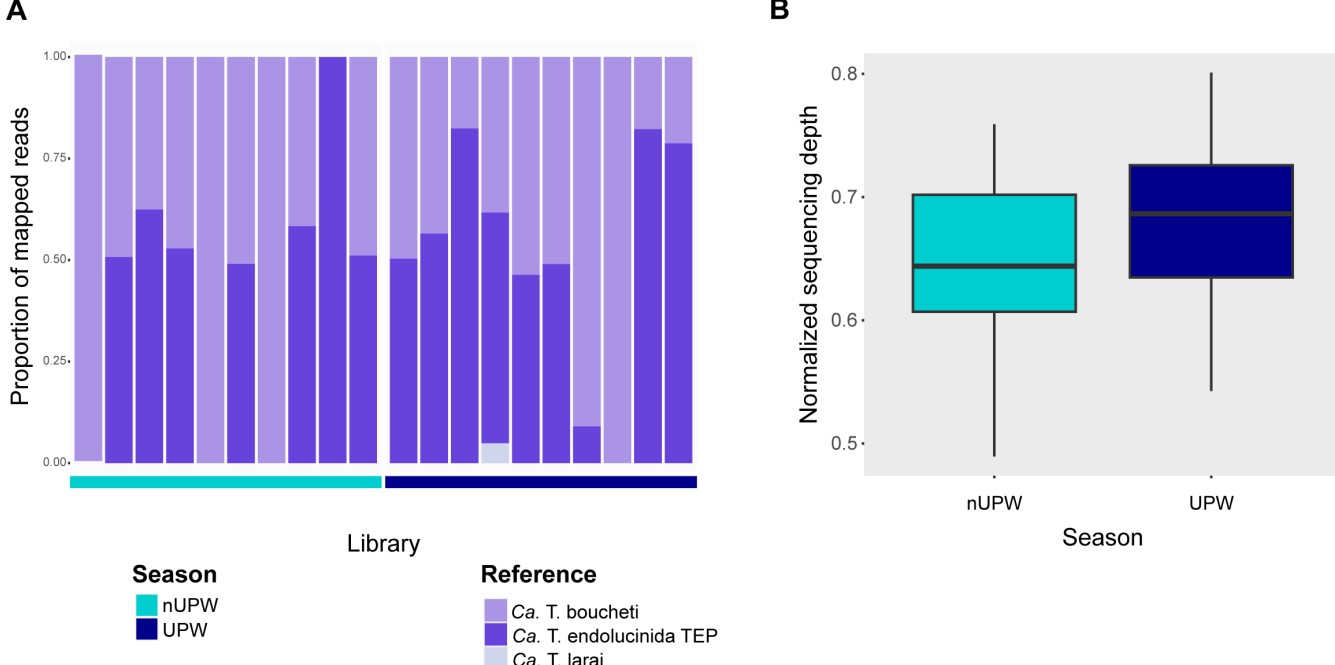

**FIG 2** Abundance and composition of primary symbionts remained stable across seasons. (A) *Ca.* T. boucheti and *Ca.* T. endolucinida TEP dominate the symbiont community in both seasons. Proportion of metagenomic reads that mapped to the corresponding symbiont reference MAG in each library. (B) Total symbiont abundance did not change significantly between seasons. Total sequencing depth of the symbionts' genomes (sum of sequencing depth of the three symbionts' reference MAGs) normalized by total microbial sequencing depth. The boxplot represents the median (center line), the lower and upper quartiles (box limits), and the minimum and maximum values (whiskers).

## Core symbiotic genes are shared by both symbiont clades and are consistently highly transcribed

We investigated the symbionts' core metabolism by characterizing the functions shared between the symbiont clades and their transcription patterns between the seasons. We compared the baseline transcription (gene median > 10 tpm) patterns of the orthogroups shared by the two symbiont clades. Most of the shared orthogroups (2,937 out of 3,436) were transcribed at both seasons by both symbiont clades (Fig. S1A). Nevertheless, we observed that each clade consistently transcribed a unique set of orthogroups across both seasons (Supplemental text and Fig. S1A). Only 55 of the shared orthogroups were among the top 100 most highly transcribed ones for each symbiont clade, regardless of season (Supplemental text and Fig. S1B). These included genes encoding a carbon storage regulator linked to biofilm formation (*csrA*), a small heat shock protein (*hspA*), a cold shock protein (*cspA*), the type VI secretion system, and a host factor protein (Supplemental text). Genes required for chemosynthetic metabolism through sulfide and methanol oxidation, including genes encoding the RubisCo enzymes, methanol dehydrogenase, and a soxYZ fusion protein, were also among the shared most highly transcribed orthogroups during both seasons. Furthermore, the cytochrome c oxidase genes were also highly transcribed, indicating that the symbionts used oxygen as the final electron acceptor regardless of the sampling time. By contrast, we found orthogroups that were among the highly transcribed exclusively during one of the compared seasons. Five genes involved in methanol oxidation (*mxaD*, *fdhA*, *MJ1427*, *xoxG*, and *mch*) were ranked among the top 100 most highly transcribed genes in both *Ca.* T. boucheti and *Ca.* T. endolucinida TEP exclusively during the non-upwelling season (Table 1). Conversely, two genes associated with the rDSR pathway (*aprA* and *aprB*) were in the top 100 transcribed genes of both clades exclusively during upwelling (Table 1).

**TABLE 1** Orthogroups with the highest transcription levels during upwelling (UPW) and non-upwelling (nUPW) seasons.[a]

| Orthogroup ID | Season | Description | Gene symbol | Ca. T. boucheti UPW (median TPM) | Ca. T. endolucinida UPW (median TPM) | Ca. T. boucheti nUPW (median TPM) | Ca. T. endolucinida nUPW (median TPM) |
|---|---|---|---|---|---|---|---|
| GC_00000510 | UPW | Cytochrome c oxidase assembly protein subunit 11 | ctaG | 2,705.09 | 3,270.23 | 927.27 | 1,248.02 |
| GC_00000520 | UPW | Anaerobic dimethyl sulfoxide reductase subunit B | dmsB | 2,552.69 | 3,540.06 | 479.44 | 1,248.1 |
| GC_00000860 | UPW | Putative dimethyl sulfoxide reductase chaperone | dmsD | 2,149.36 | 2,968.67 | 412.26 | 1,030.53 |
| GC_00000985 | UPW | Glyceraldehyde 3-phosphate dehydrogenase (phosphorylating) | gapA | 3,917.46 | 2,569.98 | 1,056.13 | 628.47 |
| GC_00001299 | UPW | Adenylylsulfate reductase, subunit A | aprA | 4,539.43 | 5,193.79 | 738.91 | 924.82 |
| GC_00001624 | UPW | Response regulator receiver domain | | 2,422.8 | 2,113.49 | 1,272.76 | 765.44 |
| GC_00002575 | UPW | Phasin protein | | 2,796.13 | 2,235.11 | 1,076.02 | 556.45 |
| GC_00002691 | UPW | Ubiquinone biosynthesis protein COQ9 | COQ9 | 2,498.94 | 1,706.565 | 1,477.02 | 1,183.33 |
| GC_00003201 | UPW | Adenylylsulfate reductase, subunit B | aprB | 5,494.25 | 3,348.8 | 1,249.14 | 693.61 |
| GC_00003256 | UPW | Hypothetical protein | | 1,627.32 | 2,012.605 | 1,213.52 | 1,110.23 |
| GC_00000259 | nUPW | Sigma-E factor negative regulatory protein | rseA | 520.79 | 653.005 | 1,869.46 | 1,649.67 |
| GC_00000354 | nUPW | Formate dehydrogenase major subunit | fdhA | 828.25 | 1,259.995 | 2,262.2 | 2,371.78 |
| GC_00000623 | nUPW | Alcohol dehydrogenase (cytochrome c) | exaA | 994.78 | 962.43 | 1,998.54 | 1,966.48 |
| GC_00000691 | nUPW | Porin | | 1,028.84 | 936.23 | 7,487.1 | 6,748.19 |
| GC_00001496 | nUPW | Beta-ribofuranosylaminobenzene 5'-phosphate synthase | MJ1427 | 300.41 | 276.22 | 3,216.16 | 3,723.94 |
| GC_00001772 | nUPW | Cytochrome c XoxG | xoxG | 725.66 | 1,527.775 | 2,694.17 | 1,714.89 |
| GC_00001918 | nUPW | MxaD protein | mxaD | 103.6 | 141.815 | 1,801.4 | 2,337.36 |
| GC_00002116 | nUPW | Methylene-tetrahydromethanopterin dehydrogenase | mtd | 345.05 | 257.365 | 2,817.58 | 1,900.89 |
| GC_00002472 | nUPW | Large subunit ribosomal protein L13 | rplM | 1,139.81 | 1,220.34 | 1,669.44 | 3,205 |
| GC_00002574 | nUPW | Hypothetical protein | | 1,247.91 | 1,401.88 | 2,305.57 | 1,829.03 |
| GC_00002631 | nUPW | Hypothetical protein | | 249.64 | 197.375 | 3,307.9 | 3,139.89 |
| GC_00003177 | nUPW | RNA polymerase sigma-70 factor, ECF subfamily | rpoE | 1,007.94 | 1,108.08 | 2,383.09 | 3,296.78 |

[a]Orthogroups ranked among the top 100 highest transcribed in the different sampling seasons by both symbiont clades and the median TPM values for the season-clade combinations.

## Energy conservation pathways are differentially expressed between upwelling and non-upwelling seasons

We analyzed the metatranscriptomes of 10 lucinids sampled in the non-upwelling and upwelling seasons, respectively (metadata and statistics in Table S2), to investigate the impact of upwelling on symbiont metabolism. Principal component analysis (PCA) of the transcriptomes of both *Ca.* T. boucheti and *Ca.* T. endolucinida TEP revealed a seasonal clustering pattern for most samples (Fig. 3A and B). Permutational multivariate analyses of variance (PERMANOVA) indicated that season and symbiont relative abundance significantly influenced transcriptional variation in *Ca.* T. boucheti (season: $R^2$ =0.087, $F = 1.59$, $P = 0.010$; relative abundance: $R^2$ =0.087, $F = 1.59$, $P = 0.030$), while total symbiont abundance was not significant ($P = 0.339$). By contrast, for *Ca.* T. endolucinida, only seasonality showed a statistically significant effect ($R^2$ =0.116, $F = 1.79$, $P = 0.025$), whereas relative abundance and total symbiont abundance were not significant ($P = 0.325$ and $P = 0.088$, respectively). Two *Ca.* T. boucheti libraries (C and S) and five *Ca.* T. endolucinida libraries (F, J, L, M, and Q) had the lowest number of reads (Table S2) and were excluded from the analysis because their respective clade was undetectable in their corresponding metagenomic libraries (Fig. 2B).

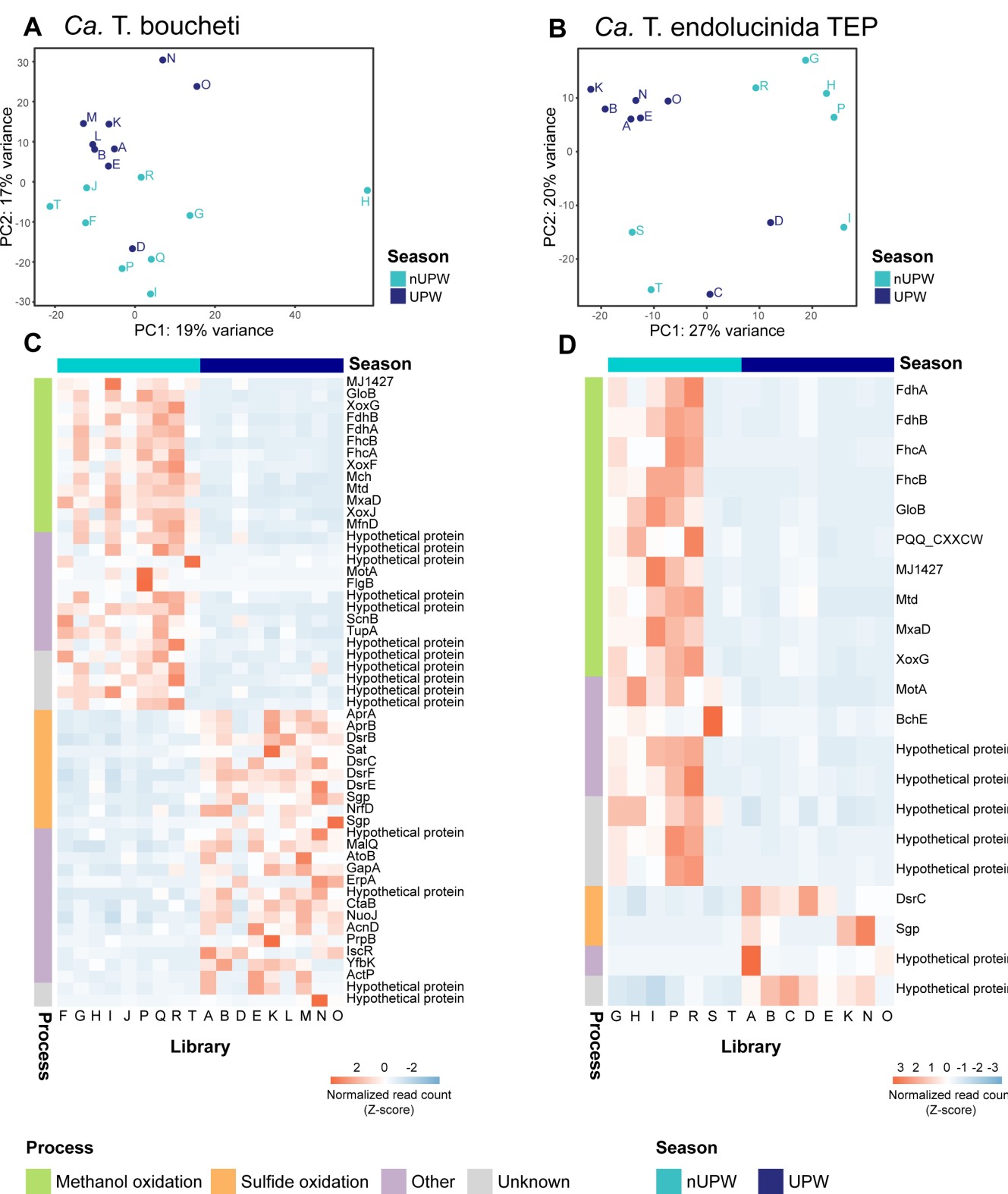

**FIG 3** Differential expression analyses revealed both symbiont clades respond to season with similar metabolic changes. (A, B) Symbiont transcriptomic patterns varied seasonally. Principal component analysis (PCA) of the transcriptomic variation of *Ca*. T. boucheti and *Ca*. T. endolucinida TEP between non-upwelling (turquoise) and upwelling (dark blue) seasons. (C, D) *Ca*. T. boucheti and *Ca*. T. endolucinida TEP upregulated genes involved in methanol oxidation and sulfide oxidation in non-upwelling and upwelling seasons, respectively. Heatmaps of Z-score normalized read counts of all genes detected as differentially expressed by DESeq2 (alpha = 0.05, lfcthreshold = 0.19). Color code represents biological processes inferred from functional annotations (Table S6).

We performed differential expression analysis to identify the symbiont genes involved in a response to upwelling-induced environmental changes. *Ca*. T. boucheti had 18 and 25 upregulated (alpha = 0.05, lfcThreshold = 0.19) genes in non-upwelling and upwelling samples, respectively, while *Ca*. T. endolucinida TEP had 17 and 4 (Fig. 3C and D; Table S6). During upwelling, *Ca*. T. boucheti upregulated seven genes involved in the sulfide oxidation pathway (rDSR) and *Ca*. T. endolucinida TEP upregulated two genes potentially involved in sulfide oxidation (Fig. 3D; Table S6).

Approximately half of the genes upregulated in non-upwelling samples, belonging to both *Ca*. T. boucheti and *Ca*. T. endolucinida TEP, were involved in methanol oxidation (Fig. 3C and D; Table S6). Comparative genomic analysis revealed a conserved gene synteny for the methanol oxidation pathway in both symbiont genomes, with the exception of a unique region in *Ca*. T. boucheti between *pqqB* and *fhcC,* containing five additional genes of unknown function (Fig. 4). This genomic region also included genes involved in the biosynthesis of tetrahydromethanopterin ($H_4$MPT) and methanofuran (MFR), cofactors essential for methanol oxidation (Fig. 4). We identified upregulated genes involved in five of the six steps of methanol oxidation to carbon dioxide in *Ca*. T. boucheti and four in *Ca*. T. endolucinida TEP (Fig. 4). These included genes for formate dehydrogenase, crucial for energy conservation, and those involved in $H_4$MPT and MFR biosynthesis (Fig. 3C, D and 4). Six *Ca*. T. boucheti genes and four *Ca*. T. endolucinida TEP genes of unknown function were upregulated in non-upwelling samples and located within the methanol oxidation gene cluster (Fig. 4), suggesting potential co-regulation and involvement in methanol oxidation. Furthermore, genes encoding the potential for formaldehyde assimilation via the serine pathway were also present in the MAGs of *Ca*. T. boucheti and *Ca*. T. endolucinida TEP. However, these genes for formaldehyde assimilation were not differentially expressed (Table S7).

## DISCUSSION

We leveraged a seasonal upwelling event in the TEP as a natural experiment to examine how lucinid symbionts respond to changing environmental conditions. Our metagenomic data indicated both *Ca*. T. endolucinida TEP and *Ca*. T. boucheti were present in the gills of *C*. cf. *galapagana*, confirming previous reports of their associations with this lucinid species in the Guanacaste region (15). Shifts in the relative abundance of symbiont types or strains, a process known as symbiont shuffling, represent a potential mechanism of microbiome-mediated acclimatization (2, 4). This process could allow symbionts better adapted to specific environmental conditions to take on a more prominent role in the symbiosis. For example, symbiont shuffling in the gill microbial community of the bivalve *Argopecten purpuratus* has been associated with changes in abiotic parameters during periods of intensified upwelling (40, 41). However, the stable composition and abundance of the *C. galapana* primary symbiont community across the two seasons (Fig. 2) suggest the two dominant symbiont species in Santa Elena Bay are functionally equivalent and well-adapted to dynamic upwelling conditions. This functional redundancy was reflected at the transcriptional level. Most of the shared highly transcribed orthogroups were consistent with aerobic chemolithoautotrophic metabolism and host-associated lifestyle (42, 43) (Supplemental text). These genes were also among the most highly expressed genes in other *Ca*. Thiodiazotropha symbionts associated with different lucinids (14, 27), likely representing the core biological functions of *Ca*. Thiodiazotropha in symbiosis. Furthermore, genes conferring clade-specific functions were not differentially expressed across seasons (Supplemental text), suggesting that the genomic differences between the clades were not involved in the response to the seasonal environmental changes. Symbiont shuffling is unlikely to be critical in the responses of this lucinid symbiosis to upwelling-induced environmental changes. Instead, association with a select few, yet metabolically versatile partners may represent an alternative strategy providing the flexibility necessary for acclimatization in lucinids.

We observed transcriptomic shifts between the upwelling and non-upwelling seasons in both *Ca*. T. endolucinida TEP and *Ca*. T. boucheti that could not be attributed to

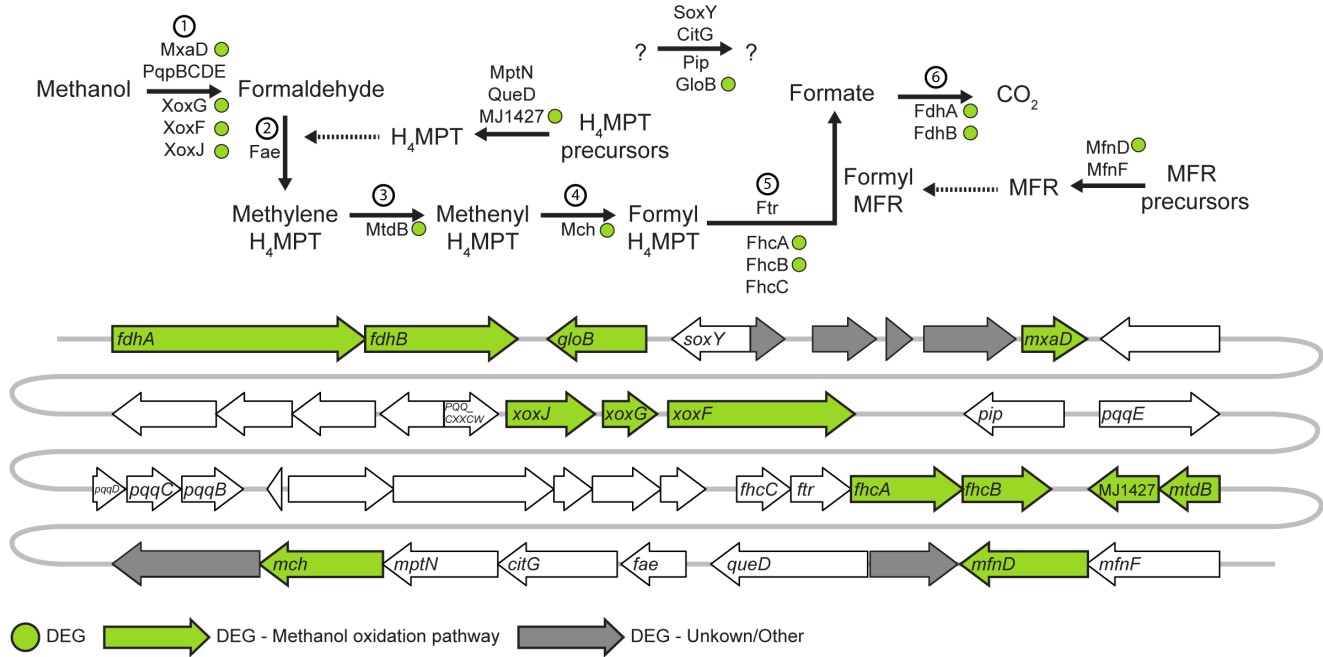

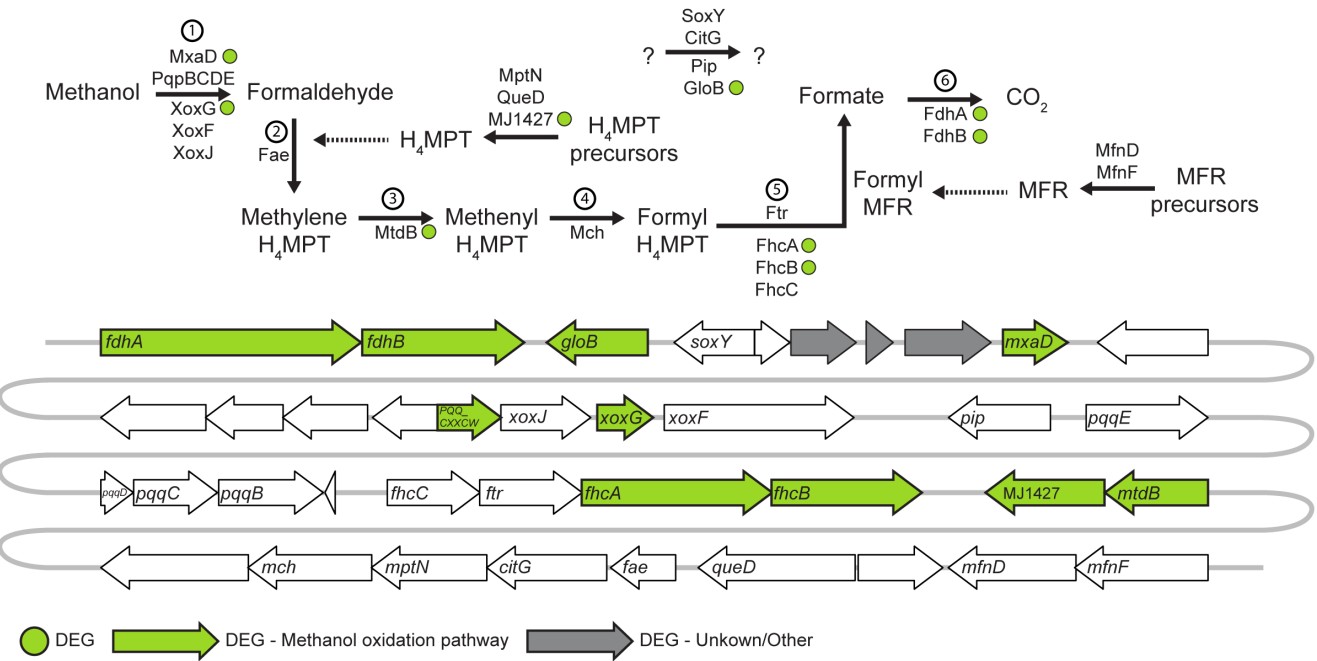

**FIG 4** The dissimilatory methanol oxidation pathway is upregulated in the non-upwelling season and conserved across symbiont clades. Gene organization and differentially expressed genes (DEG) in (A) *Ca*. T. boucheti and (B) *Ca*. T. endolucinida TEP.

changes in the total abundance of both species. By contrast, the season as well as the relative abundance of *Ca*. T. boucheti, but not *Ca*. T. endolucinida TEP, had a statistically significant effect on global transcriptomic variation, although these factors only

accounted for less than 30% of the total variance observed, indicating that additional factors (e.g., host physiology) may have played a substantial role. Further research is needed to explore these contributing factors. Both symbiont clades responded to seasonal changes with shifts in the transcription levels of genes involved in energy conservation pathways. Specifically, genes involved in sulfur oxidation were upregulated during the upwelling season (March), while genes involved in methanol oxidation were upregulated during the non-upwelling season (November), which may indicate that sulfide oxidation played a more significant role in the symbionts' energy budget during upwelling, while methanol oxidation was more important during non-upwelling periods.

Our upwelling sampling coincided with the end of the Papagayo upwelling event in March (Fig. 1B), and the observed changes in primary productivity and abiotic conditions were consistent with the typical timing, duration, and effects of this event, which occurs between December and April in the Guanacaste region (32, 33, 44, 45). Globally, patterns of sulfate reduction often reflect primary productivity and phytoplankton biomass, particularly in regions such as the eastern equatorial Pacific, where elevated rates of sulfate reduction occur beneath nutrient-rich upwelling zones (46). Upwelling increases the flux of organic matter to the seafloor, promotes aerobic respiration, depletes oxygen, and creates conditions favorable for sulfate reduction and thus, the production of sulfide (47–49). While studies on the effects of seasonal upwelling in Santa Elena Bay are limited, previous work has reported strong variability in seawater physico-chemical parameters and enhanced respiration in the water in response to these events (50). In the neighbouring Gulf of Papagayo, upwelling influences both surface and bottom water dynamics, with peak chlorophyll concentrations coinciding with minimum oxygen levels (51). In nearby Golfo Dulce, increased abundances of sulfur-oxidizing bacteria and high sulfide oxidation rates during the dry season are consistent with the upwelling-induced increase in organic matter (52). These regional trends lead us to propose that there is increased production of reduced sulfur compounds in Santa Elena Bay during the upwelling season, which could explain the upregulation of genes involved in the rDSR pathway by lucinid symbionts (Fig. 3C and B).

By contrast, lucinids sampled in November (non-upwelling) likely experienced prolonged low primary productivity in Santa Elena Bay (Fig. 1B). Methanol production in the marine environment, particularly in the sediments, remains poorly understood (53, 54). However, methanol concentrations in the water column are inversely correlated with primary productivity (55), while microbial methanol uptake is positively correlated with phytoplankton abundance (56), which could lead to increased methanol availability during periods of low productivity. In addition, known sources of methanol include methane oxidation via methane monooxygenase and anaerobic decomposition of organic matter (53), both of which can be inhibited by sulfide (57, 58). We speculate that the reduced organic matter deposition during non-upwelling conditions results in sulfide limitation and increased methanol availability, potentially due to lower methanol turnover. These factors could favor increased reliance on methanol as an alternative electron donor during non-upwelling periods (Fig. 3B and C). Although we detected the potential for formaldehyde assimilation via the serine pathway from methanol oxidation—previously reported in other lucinid symbiont clades (14)—these genes were not differentially expressed, suggesting that methanol was primarily used as an energy source rather than for carbon assimilation. Furthermore, the consistently high RubisCO gene transcription levels across both seasons (Supplemental text and Fig. S2) suggest that the symbionts were able to sustain carbon fixation under changing environmental conditions. We hypothesize that the changes in the transcription of symbiont methanol and sulfur oxidation pathways reflect seasonal shifts in energy resource availability, driven by changes in sediment biogeochemistry and microbial activity associated with upwelling dynamics. However, upwelling also alters physicochemical parameters of the water column, such as temperature, salinity, and pH, all of which could influence both the availability of electron donors in the sediment and the efficiency of the different oxidation pathways. To support this hypothesis and tease apart the specific effects of the

Papagayo upwelling on sediment biogeochemistry, additional seasonal measurements of environmental parameters, along with controlled experiments quantifying rates of sulfur and methanol oxidation and carbon fixation, are needed.

The methanol oxidation gene cluster is widespread across the genus *Ca*. Thiodiazotropha (14, 25). The conserved synteny of this genomic region in distantly related *Ca*. Thiodiazotropha associated with *Ctena orbiculata* (27) and further suggests that this pathway may be essential for the resilience of this symbiont clade in diverse environmental contexts. However, previous studies have reported low to moderate transcription and expression levels of this pathway (14, 27, 28) in lucinid symbionts from seagrass meadows and the deep sea, suggesting a potential accessory role in less variable environments. The high transcription of the dissimilatory methanol oxidation genes (Table 1) in our non-upwelling samples reveals the potential ecological importance of the pathway and lucinid symbiont metabolic flexibility in highly variable environments, such as the upwelling-influenced rocky intertidal zone of Santa Elena Bay. Conversely, some lucinid symbionts, including the Mediterranean *Ca*. T. weberae and lotti, and symbionts of the genus *Sedimenticola* (14, 25) lack methanol oxidation genes and are unable to use methanol as an alternative energy source when reduced sulfur compounds are limited. Carbon fixation rates were observed to decrease in *Loripes orbiculatus* symbionts under sulfide-poor conditions in the Mediterranean Sea (59). Further investigation is required to determine if there is a causal relationship between the observed decrease in carbon fixation rates and the inability of these symbionts to utilize methanol as an electron and energy source. Nevertheless, these results could be an indication of different physiological responses and ecological adaptations to environmental change in the coastal environments of lucinid symbionts, linked to their diverse genomic capabilities. This variability in the responses of different symbiont clades to fluctuations in resource availability underscores the importance of considering the response of host-associated microbes when predicting ecological outcomes of stressors such as anthropogenic changes in nutrient regimes of coastal environments.

## MATERIALS AND METHODS

### Sample collection

Ten specimens of *Ctena* cf. *galapagana* were collected on 4 November 2022 (non-upwelling) and another 10 were collected on 21 March 2023 (upwelling) at Santa Elena Bay, Guanacaste, Costa Rica (10.921,–85.809488; Table S1). Clams were dug up from the low intertidal zone at ~30 cm sediment depth during low tide, opened on-site with a razor blade, and the entire soft tissues were preserved in DNA/RNA Shield (catalog number R1100-250; ZymoBiomics) according to the manufacturer's instructions. Samples were kept at room temperature during transport and stored for long term at −20°C.

### Environmental data

To investigate the effects of the upwelling on environmental conditions within Santa Elena Bay, data were extracted from the publicly available database Copernicus MyOcean Pro (https://data.marine.copernicus.eu/) by creating a polygon of points encompassing the water in the bay (coordinates: −85.8094765223726, 10.932366577689425; −85.81222310440386, 10.92206689507224; −85.80055013077104, 10.91726037651755; −85.79162373916947, 10.917947022025364; −85.8094765223726, 10.932366577689425). Daily temperature, chlorophyll A, nitrate, and oxygen concentrations were extracted for the window of 15 April 2022, to 15 September 2023, and then plotted in R using ggplot2 v3.5.1 (60). All measurements correspond to a 5 m depth. Maps were plotted using rnaturalearth v1.0.1 (61) and ggspatial v1.1.9 (62).

## Nucleic acid extraction and sequencing

### *Metatranscriptomes*

Soft tissues of the specimens were placed in bead-beater lysis tubes containing sterile silicone beads and TRIzol (catalog number 15596018, Thermo Fisher Scientific). The tissues were homogenized during two consecutive steps of bead beating at 4.0 m/s for 10 seconds each in a FastPrep instrument (MP Biomedical). Total RNA was extracted by phase separation according to the TRIzol manufacturer's protocol. RNA extractions were treated with TURBO DNase (catalog number AM2238, Invitrogen) to deplete any residual DNA, and then purified using the GeneJET RNA Purification Kit (catalog number K0731, Thermo Fisher Scientific). To enhance the sequencing of symbiont mRNA and thus facilitate the quantification of the transcription of coding genes, both bacterial RNA and host ribosomal RNA (rRNA) were depleted in accordance with the manufacturer's instructions using the general Pan-Bacteria riboPOOL kit (kit dp-P024-000026, siTOOLs Biotech), and a custom-designed host-specific riboPOOL kit (dp-K024-000089, siTOOLs Biotech), which was created using the various lucinid rRNA sequences (detailed methods in Supplemental text; sequences available on FigShare [63]). Subsequently, libraries from each clam were prepared with the NEBNext Ultra II RNA Library Prep Kit for Illumina (catalog number E7770, New England Biolabs), including RNA fragmentation. Subsequent Illumina sequencing was conducted on a NextSeq 2000 at the Max Planck Genome-Centre Cologne to generate a minimum of 50,000,000 2 × 150 bp reads per sample library (Table S2).

### *Metagenomes*

DNA was isolated from the remaining interphase and organic phase of the Trizol RNA extraction according to an adapted version of the manufacturer's protocol described in reference (25). The extracted DNA was purified with the CleanAll DNA/RNA Clean-Up and Concentration Micro Kit (catalog number 23800, Norgen Biotek), and sequencing libraries from each clam were constructed using the Nextera LITE DNA library preparation kit (64). Illumina sequencing was conducted on the NextSeq 2000 platform at the Max Planck Genome-Centre Cologne to produce at least 14,000,000 2 × 150 bp paired-end reads per library (Table S2).

## Symbiont composition and abundance

### *Quality filtering, assembly, and binning*

Following the removal of adapters and PhiX contamination, metagenomic reads were trimmed and filtered to a minimum length of 100 bp using the BBDuk feature of BBMap v39.00 (65). Individual read libraries were assembled using SPAdes v3.9.1 (66) (--meta, -k 21,31,41,51,61,71,81,91). The resulting contigs were filtered to a minimum length of 1,000 bp and then binned with MetaBAT2 v2.16 (67), Binsanity v0.5.4 (68), and MaxBin2 v2.2.6 (69). The metagenome-assembled genome (MAG) bins were aggregated and dereplicated using DAS Tool v1.1.6 (70). The differential coverage data necessary for binning were obtained with BWA-MEM v0.7.17 (71), SAMtools v1.19.2 (72), and the pileup.sh script from BBMap v39.00 (65). MAG quality was assessed using CheckM2 v1.0.1 (73), and the taxonomy of the high-quality MAGs (>90% completeness and <5% contamination) was assigned by calculating the ANI, using fastANI v1.33 (74), against previously published *C.* cf. *galapagana* symbiont MAGs recovered from Santa Elena Bay (15).

### *Breadth and depth of coverage*

Reference genomes were selected from both newly recovered and previously published MAGs from Santa Elena Bay, based on genome completeness, contamination, and contiguity (highlighted in Table S3). Selection prioritized species previously reported to be associated with *Ctena galapagana*, as well as those identified through competitive

read mapping using CoverM v0.6.1 (75) (-p bwa-mem --trim-min 10 --trim-max 90 --min-read-percent-identity-pair 0.96 --min-read-aligned-percent-pair 0.9). Metagenomic libraries were mapped against a comprehensive set of MAGs representing all currently described *Ca*. Thiodiazotropha symbionts (Table S5). Symbiont species that exhibited a breadth of coverage greater than 15% in any library were retained as references, excepting the clade "CTENA4," a lineage reported only in the Caribbean (14, 15), which shares >95% ANI with *Ca*. T. endolucinida TEP. Due to its low coverage and high similarity, this likely was a product of cross-mapping, and the species was excluded from the final reference set. The breadth of coverage and the truncated average depth of coverage (TAD80) were determined by competitive mapping of the trimmed reads to reference MAGs of *Ca*. Thiodiazotropha boucheti, *Ca*. T. endolucinida TEP and *Ca*. T. larai using CoverM v0.6.1 (75) with stringent parameters to minimize cross-mapping between *Ca*. T. boucheti and *Ca*. T. endolucinida TEP, which exhibits an average ANI of 94%. (-p bwa-mem --trim-min 10 --trim-max 90 --min-read-percent-identity-pair 0.98 --min-read-aligned-percent-pair 0.9). To visualize the relative abundance of each symbiont clade, a symbiont clade was identified as present in the sample if the breadth of coverage of its corresponding reference MAG was greater than 15%. The TAD80 of each MAG within a library was relativized by the sum of the TAD80 values per library. Statistical analyses and plots were done in R using the ggplot2 v3.5.1 package (60). The sum of the TAD80 values of the three reference MAGs within a given library (total symbiont sequencing depth) was normalized by the "genome equivalent" (i.e., total sequencing depth of microbial genomes) of the corresponding library, which was calculated from the trimmed reads using MicrobeCensus v1.1 (76). This quotient was used as a proxy for symbiont abundance (total normalized sequencing depth of the symbionts' genomes, TNSD), as described in reference 39. Normality and homogeneity of variance of the TNSD were evaluated with a Shapiro test and a Levene test, respectively, and a two-way Student's *t*-test with a confidence level of 0.95 was performed to test for statistically significant differences in symbiont abundance between the upwelling and non-upwelling groups (Table S5).

## Metatranscriptome analysis

### Splitting and mapping reads

Trimming of the metatranscriptomic reads was done as described for the metagenomic reads. rRNA reads were identified and filtered out using SortMeRNA v4.3.4 (77) by aligning reads against the sensitive database and a custom database containing lucinid host and symbiont rRNA sequences (detailed methods in Supplemental text; sequences available on FigShare [78]). The remaining reads were assigned to their corresponding symbiont clade by competitive mapping against reference MAGs (See "Breadth and depth of coverage") of *Ca*. Thiodiazotropha boucheti, *Ca*. T. endolucinida TEP, and *Ca*. T. larai using the BBSplit feature of BBMap v39.00 (65) (minid = 0.98, slow = t, ambiguous = best, ambiguous2 = toss, pairedonly = t). To obtain the mapping coordinates, the split reads were re-mapped against their corresponding reference MAG with BBMap v39.00 (65). Given the low number of reads mapping to the reference MAG of *Ca*. T. larai in most libraries, it was excluded from subsequent transcriptomic analyses.

### Read counts

The reference MAGs were annotated with Prodigal v2.6.3 (79) implemented through Anvi'o v8 (80), by generating contigs databases with "anvi-gen-contigs-database," and the gene models were subsequently functionally annotated using the KOfam HMM database of KEGG orthogroups (81, 82) with "anvi-run-kegg-kofams" (83). In addition, the predicted genes were functionally annotated with eggnog-mapper v2.1.12 (--itype CDS -m diamond) (84–86) against the eggNOG database (5.0.2). A GFF3 file was obtained from the contigs-database of each reference with "anvi-get-sequences-for-gene-calls" (with --export-gff3 flag), and the mapping positions obtained from BBMap were used as

input to produce raw read counts and transcripts per million (TPM) values using FADU v1.9 (--remove_multimapped) (87).

## Differential gene expression

All differential gene expression analyses were performed independently for *Ca.* T. boucheti and *Ca.* T. endolucinida TEP using the DESeq2 v1.42.0 R package (88). Libraries with less than 20,000 reads mapped with BBSplit (Table S2) and with no detection of the symbiont clade in the corresponding metagenomic library were excluded from all further RNA-Seq analyses.

After filtering out genes with less than five counts across libraries and outlier removal, the raw counts were normalized via VST, and PCA plots were generated to visualize sample clustering. The contribution of season, symbiont abundance (TNSD), and relative proportion of symbionts to transcriptional variation was assessed by permutational multivariate analysis of variance (PERMANOVA) on the Euclidean distances calculated from the VST data, using the adonis2 function from the vegan v2.6.6.1 R package (89). The homogeneity assumption of the multivariate dispersion was tested using the betadisper and permutest functions from the same package. Differentially expressed genes (DEGs) between seasons were identified by running the default DSEq function on the raw counts and using independent hypothesis weighting (90) (alpha = 0.05, lfcThreshold = 0.19). $\log_2$ fold change adaptive shrinkage estimation was done via ashr (91). Genes with an adjusted *P*-value < 0.05 were considered to be differentially expressed.

## Comparative transcriptomics

A pangenome database was computed from the annotated contigs databases (see section Read Counts) of new and publicly available MAGs of *Ca.* T. boucheti and endolucinida TEP (Table S3) using "anvi-pan-genome" (Anvi'o v8 (80, 92)). The assignment of predicted genes to Anvi'o pangenome's gene clusters, here referred to as orthogroups, of the reference MAGs was extracted from the pangenome database using the "anvi-export-table" function. The transcript per million (TPM) values of all genes within a given orthogroup were aggregated for each sample. The median TPM of orthogroups per combination of season and clade was estimated using the R package tidyverse v2.0.0 (93), and orthogroups with a median TPM above 10 were considered to be baseline-transcribed. Furthermore, the top 100 orthogroups with the highest median TPM values were extracted for each clade-season combination. A set-based analysis was performed to identify the transcribed orthogroups of *Ca.* T. boucheti and *Ca.* T. endolucinida TEP across the different seasons. Upset plots were obtained from both the baseline and most highly transcribed orthogroups with the R package ComplexUpset v1.3.3 (94) to visualize the intersections across season-clade combinations. The identity of the orthogroups within each intersection was extracted in R.

## ACKNOWLEDGMENTS

We are grateful to Minor Lara, Jr., Steven Lara, and their mother Ivannia, Jonathan Cybulski, Teresa Peil, Kevelyn Vargas, Cristopher Campos, Ennio Schmid, and Kimberly García-Méndez for their participation during fieldwork in Guanacaste. We thank Dr. Bruno Huettel at the Max Planck Genome Centre Cologne for his support in library preparation and sequencing. We would also like to thank Dr. Tim Ferdelman and Dr. Dirk de Beer for their helpful input in interpreting the sediment biogeochemistry of the system, and Dr. Manuel Kleiner for his valuable insights into symbiont physiology. Finally, we would like to acknowledge the use of ChatGPT, an AI language model, for its assistance in optimizing the coding of the bioinformatics pipeline. Collection in Costa Rica and following export and sequencing were performed under permits R-004-2019-OT-CONAGEBIO and R-017-2022-OT-CONAGEBIO from the Comisión Nacional para la Gestión de la Biodiversidad (CONAGEBIO).

This project was funded by the Max-Planck-Gesellschaft. A.C.K., B.Y., L.G.W.E., and I.M.-L. received salaries from Max-Planck-Gesellschaft. L.G.E.W. was supported with salary and a postdoctoral fellowship by a Marie Curie individual postdoctoral fellowship "MSCA-IF-EF- RI" for project #Pansymbiosis with grant number SEP-210693430.

## AUTHOR AFFILIATIONS

[1]Eco-Evolutionary Interactions Group, Max-Planck-Institute for Marine Microbiology, Bremen, Germany

[2]Department of Molecular Ecology, Max-Planck-Institute for Marine Microbiology, Bremen, Germany

[3]Centro de Investigación en Ciencias del Mar y Limnología (CIMAR), Universidad de Costa Rica, San José, Costa Rica

[4]Centro de Investigación en Biodiversidad y Ecología Tropical (CIBET), Universidad de Costa Rica, San José, Costa Rica

[5]Escuela de Biología, Universidad de Costa Rica, San José, Costa Rica

[6]Diving Center Cuajiniquil, Provincia de Guanacaste, Cuajiniquil, Costa Rica

[7]Smithsonian Tropical Research Institute, Balboa, Ancon, Panama

## AUTHOR ORCIDs

Isidora Morel-Letelier http://orcid.org/0000-0003-1402-2393
Laetitia G. E. Wilkins http://orcid.org/0000-0003-3632-2063

## FUNDING

| Funder | Grant(s) | Author(s) |
| --- | --- | --- |
| Max-Planck-Gesellschaft | | Isidora Morel-Letelier |
| H2020 Marie Skłodowska-Curie Actions | SEP-210693430 | Laetitia G. E. Wilkins |

## AUTHOR CONTRIBUTIONS

Isidora Morel-Letelier, Conceptualization, Data curation, Formal analysis, Investigation, Methodology, Project administration, Validation, Visualization, Writing – original draft, Writing – review and editing | Benedict Yuen, Conceptualization, Data curation, Formal analysis, Investigation, Methodology, Project administration, Supervision, Validation, Writing – original draft, Writing – review and editing | Luis H. Orellana, Conceptualization, Data curation, Formal analysis, Investigation, Methodology, Project administration, Supervision, Validation, Writing – original draft, Writing – review and editing | A. Carlotta Kück, Conceptualization, Formal analysis, Investigation, Methodology, Supervision, Writing – review and editing | Yolanda E. Camacho-García, Conceptualization, Resources, Writing – review and editing | Minor Lara, Investigation, Methodology, Resources | Matthieu Leray, Conceptualization, Resources, Writing – review and editing | Laetitia G. E. Wilkins, Conceptualization, Data curation, Funding acquisition, Investigation, Methodology, Project administration, Supervision, Validation, Writing – original draft, Writing – review and editing

## DATA AVAILABILITY

Raw read sets were deposited in the NCBI BioProject PRJNA1176339 with Biosample accession numbers SAMN44399338–SAMN44399357. High-quality MAGs retrieved in this study were deposited in the same BioProject with accession numbers SAMN44405716–SAMN44405718. All the scripts and a detailed description of all bioinformatic and statistical analyses performed can be found in the publicly available GitLab repository at https://gitlab.mpi-bremen.de/imorel/riding_the_upwelling. Additional data sets were made available on Figshare, https://figshare.com/projects/Riding_the_upwelling/227328.

## ADDITIONAL FILES

The following material is available online.

### Supplemental Material

**Supplemental figures (mSystems01686-24-S0001.docx).** Figures S1 and S2.
**Supplemental text (mSystems01686-24-S0002.docx).** Additional methods, results, and discussion.
**Table S1 (mSystems01686-24-S0003.xlsx).** Monthly means and daily satellite measurements.
**Table S2 (mSystems01686-24-S0004.xlsx).** Metadata of samples, metagenomes, and metatranscriptomes.
**Table S3 (mSystems01686-24-S0005.xlsx).** Metadata of MAGs.
**Table S4 (mSystems01686-24-S0006.xlsx).** ANI values of MAGs.
**Table S5 (mSystems01686-24-S0007.xlsx).** Sequencing depth of reference MAGs.
**Table S6 (mSystems01686-24-S0008.xlsx).** *Ca*. T. boucheti and endolucinida TEP's differentially expressed genes.
**Table S7 (mSystems01686-24-S0009.xlsx).** Differential expression analysis results of the serine cycle.

### Open Peer Review

**PEER REVIEW HISTORY (review-history.pdf).** An accounting of the reviewer comments and feedback.

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
