## [Reviewer comments · mSystems]

Seasonal transcriptomic shifts reveal metabolic flexibility of chemosynthetic symbionts in an upwelling region

Isidora Morel Letelier, Benedict Yuen, Luis Orellana, A. Carlotta Kück, Yolanda Camacho-García, Minor Lara, Matthieu Leray, and Laetitia Wilkins

Corresponding Author(s): Isidora Morel Letelier, Max Planck Institute for Marine Microbiology

Review Timeline:

Submission Date:	December 12, 2024
Editorial Decision:	March 31, 2025
Revision Received:	April 21, 2025
Accepted:	April 25, 2025

Editor: Hans Bernstein

Reviewer(s): The reviewers have opted to remain anonymous.

Transaction Report:

DOI: <https://doi.org/10.1128/msystems.01686-24>

Re: mSystems01686-24 (Transcriptomic shifts reveal metabolic flexibility of chemosynthetic symbionts in response to upwelling-induced environmental change)

Dear Dr. Isidora Morel Letelier:

First, thank you for being patient during this review process and I apologize that it took a bit longer. I was only able to obtain one quality peer review (see below) and decided not to wait any longer on reviewers #1 & 3. I therefore reviewed this article myself and am copying my comments and suggestions for improvement below. Overall, the reviewer and I found this paper to be of high quality. It is well written and of high scientific interest. There are several suggested edits that I hope you will respond to by making changes to the manuscript by paying specific attention to the comments about making a better link between environmental changes and the observed transcriptomic shifts.

Revision Guidelines

Sincerely,
Hans Bernstein
Editor
mSystems

Reviewer #2 (Comments for the Author):

This study investigates how chemosynthetic symbionts in lucinid bivalves adapt metabolically to environmental changes brought

about by seasonal upwelling in the Tropical Eastern Pacific. Upwelling events in Santa Elena Bay, Costa Rica, significantly alter water temperature, oxygen levels, and nutrient availability, providing a natural experiment to study host-microbe responses to environmental shifts.

L140 The ANI values of these genomes are exceptionally high, nearly 100%. How did the authors differentiate these genomes when calculating relative abundance? If they were grouped together and CoverM was used for abundance estimation, the reliability of the results could be compromised. Could the authors elaborate on their approach to address this issue?

L167 The statement "Upwelling conditions did not alter symbiont abundance or composition" is based on MAG results. Did the authors perform read-level annotations to analyze potential differences in microbial abundance and composition? This step could be crucial, as some symbiotic microbial MAGs might not have been recovered during the analysis.

L193 & L197 Is there any available data on the concentrations of sulfide, methanol, or other relevant compounds? Such information would provide valuable context for interpreting the metabolic activity of symbionts.

L229 The authors highlight that methanol and sulfide oxidation pathways are critical for symbiont metabolism under different environmental conditions, particularly during upwelling. Have these pathways also been observed in other ecosystems, such as cold seeps or hydrothermal vent regions?

L257 As the authors note, additional physicochemical parameters are needed to strengthen the conclusions. Therefore, it is suggested to tone down the language in this subsection and in similar conclusions throughout the manuscript to reflect the current limitations.

L174 & L233 There appears to be some overlap between the third and fourth sections. It seems this may result from combining results and discussion. To improve clarity, the authors might consider separating the results and discussion into distinct sections.

L288 The samples were transported at ambient temperature. How long was the time from sampling to freezing? Could this delay have affected the subsequent metagenomic and metatranscriptomic analyses?

L319 It would be helpful if the authors explicitly reported the final sequencing output for each metatranscriptome dataset.

L328 Typically, metagenomic sequencing outputs are higher than those for metatranscriptomes. However, in this study, the opposite trend was observed. Could the authors clarify the rationale behind this choice of sequencing output?

The authors did not provide details about the biological replicates for the metatranscriptomes, which is a critical aspect.

Editorial review comments

This study investigates how chemosynthetic symbionts in *Ctena cf. galapagana* clams respond to seasonal environmental changes caused by coastal upwelling in Costa Rica. Using metagenomic and metatranscriptomic analyses, the authors compared symbiont communities and gene expression before and during the upwelling season. While the symbiont composition remained stable, both dominant symbiont clades shifted their gene expression in response to changing conditions. Methanol oxidation genes were more active before upwelling, while sulfur oxidation genes were upregulated during upwelling, reflecting changes in available energy sources. These findings highlight the metabolic flexibility of the symbionts and suggest this adaptability plays a key role in helping the host thrive in a dynamic environment.

The introduction could be improved by giving a little more context into the functional capacity and lifestyles of the symbionts that were targeted via transcriptome analysis. This would give the readers a bit more background before getting into the specifics about "upwelling-induced" differential expression of the various pathways but with emphasis on the role of methanol and sulfide oxidation.

The thing that seemed to be missing from the Introduction and early Results section were quantified measures of how the environment changed with respect to upwelling. There was very little info presented in the paper/figures with respect to environmental measurements, which could have included (for example) methanol/sulfide concentrations, temperature, pH, etc. The authors use the term "upwelling-induced" when referring to gene expression, however my view is that upwelling itself is not the driver of differential gene expression but rather the upwelling-associated changes in environment. This simple shift in organizing the Results and adding more context to the Introduction could add a lot to the interpretability of the Results.

Here are two suggestions of how the authors can better link observed changes in transcriptional behavior to environmental variability without having to run additional experiments:

i) Leverage existing environmental datasets more effectively or use other published geochemical profiles from the region to

more explicitly correlate transcriptomic shifts with changes in substrate availability (e.g., sulfide or methanol levels) during upwelling vs. non-upwelling conditions.

ii) Quantify and compare the expression levels of key metabolic genes relative to core housekeeping genes or total transcript counts to better estimate the relative contribution of different energy pathways under each condition, providing a clearer picture of symbiont metabolic adaptation as it relates to the interpretations around energy metabolism.

Other comments:

L180 "Since symbiont abundance and composition remained relatively constant between non-upwelling and upwelling samples (Fig. 2), the observed transcriptomic differences can be attributed to environmental changes, rather than factors like population size or interactions between the two symbiont clades." - since the authors have already performed PCA analysis, then it would be useful to perform a permanova analysis on this data to see if environmental change does indeed correlate more with the observed variance as compared to population size or any other measurements taken.

Minor comments:

There are several instances where the species names are not given in italics. E.g., L213

Reviewer #2 (Comments for the Author):

This study investigates how chemosynthetic symbionts in lucinid bivalves adapt metabolically to environmental changes brought about by seasonal upwelling in the Tropical Eastern Pacific. Upwelling events in Santa Elena Bay, Costa Rica, significantly alter water temperature, oxygen levels, and nutrient availability, providing a natural experiment to study host-microbe responses to environmental shifts.

We would like to extend our heartfelt gratitude to Reviewer #2 for the dedication and effort in reviewing our manuscript. We understand that reviewing manuscripts is a voluntary and time-consuming task, and we sincerely appreciate their commitment to providing thoughtful feedback. Reviewer #2's encouraging constructive feedback has contributed significantly to the improvement of the quality of our manuscript, especially in the clarity, structure, and flow. Please note that the line numbers correspond to the clean copy of the revised manuscript.

L140 The ANI values of these genomes are exceptionally high, nearly 100%. How did the authors differentiate these genomes when calculating relative abundance? If they were grouped together and CoverM was used for abundance estimation, the reliability of the results could be compromised. Could the authors elaborate on their approach to address this issue?

The ANI stated in now L151-152 refers to the values against reference MAGs, calculated to confirm the taxonomy of the newly retrieved MAGs. We understand where the confusion can arise, so this has been clarified in the text: "Two high-quality MAGs were classified as *Ca. T. boucheti* (Average Nucleotide Identity, ANI ~ 99.2% **to reference MAGs**) and one as *Ca. T. endolucinida* TEP (ANI ~ 99.8% **to reference MAGs**; S4 Table)".

Furthermore, the ANI between the *Ca. T. boucheti* and *Ca. T. endolucinida* TEP is, on average, 94% (Morel-Letelier et al. 2024), while *Ca. T. larai* is more distantly related to both of them. CoverM (which was performed under a competitive mapping strategy) should avoid cross-mapping between the two genomes, given their difference in ANI as stated above. Additionally, we included a minimum identity of 98% for the mapping with CoverM, to further ensure no cross-mapping (clarified now in Methods section, L424-427). This strategy was used for the mapping of metagenomic and metatranscriptomic reads.

L167 The statement "Upwelling conditions did not alter symbiont abundance or composition" is based on MAG results. Did the authors perform read-level annotations to analyze potential differences in microbial abundance and composition? This step could be crucial, as some symbiotic microbial MAGs might not have been recovered during the analysis.

Our analyses of symbiont abundance and composition were based on normalized read mapping to a curated set of reference MAGs classified as *Ca. T. boucheti*, *endolucinida*, and *larai* (described in the Methods section: "Breadth and depth of coverage"). We chose to use these as references because extensive regional sampling has shown that *Ctena galapagana* is consistently associated with these three symbionts (Morel-Letelier et al. 2024), which aligns with the taxonomy of the few newly recovered MAGs.

To further address the possibility that additional symbiont species may have been present but not recovered as MAGs, we conducted two complementary analyses:

First, we performed competitive mapping of all metagenomes against MAGs from all 27 described *Ca. Thiodiazotropha* species using CoverM. This analysis revealed that in almost all libraries, only *Ca. T. boucheti*, *endolucinida*, and *larai* exceeded thresholds of >1X depth and >10% coverage (i.e., on the lower detection limit). The only exception was a MAG from the "Ctena4" clade, which slightly exceeded these thresholds in only two libraries. However, this MAG was previously reported only in the Caribbean, has nitrogen fixation potential (absent in TEP symbionts (Morel-Letelier et al. 2024)), and shares a high ANI (95.4%) with *Ca. T. endolucinida*. Due to its low coverage and high similarity, we considered this likely to reflect cross-mapping and thus, we excluded it from the final reference set. This analysis is now described in detail in the revised Methods (L410-421), with results presented in S5 Table.

Second, we performed short-read taxonomic annotation using Metabuli (Kim & Steinegger 2024) with GTDB release r220, to which we added 22 *Ca. Thiodiazotropha* MAGs that were not represented originally in the GTDB. *Ca. Thiodiazotropha* consistently emerged as the most abundant bacterial genus in all metagenomes (1.3-5.8% of reads), distantly followed by *Enterococcus* (~0.01%, likely from host gut microbiota or contamination). Among the reads annotated as *Ca. Thiodiazotropha*, 74-90% corresponded to *Ca. T. boucheti*, *endolucinida*, and/or *larai*. However, some annotations were inconsistent with the CoverM mapping results. For example, *Ca. T. fergusonii* was the fourth species with the highest representation in the metabuli annotations, but had low representation in the metagenomic samples (breadth <0.4%, depth <0.4X), suggesting that these were likely false positives. Due to the lower resolution and potential noise of the read-level annotations, we decided not to include them in the manuscript.

Building on previous studies from this region, and supported by both the recovered MAGs and our short-read analyses, we are confident that our reference-based approach captures the full diversity of primary symbionts in this system. However, because our study focuses specifically on primary symbionts (and not, for example, the gut microbiome), we have revised the language throughout the manuscript to be more conservative and reflective of this scope (e.g. Abundance and composition of **primary** symbionts remained stable across seasons. (L801))

L193 & L197 Is there any available data on the concentrations of sulfide, methanol, or other relevant compounds? Such information would provide valuable context for interpreting the metabolic activity of symbionts.

We agree that having sediment geochemical profiles through time (in addition to satellite measurements of chlorophyll, nitrate and oxygen concentrations, and seawater temperature) would provide a more mechanistic understanding of the relationship between transcriptional shifts and dynamic fluctuations in substrate availability during upwelling and non-upwelling. However, to our knowledge, time series biogeochemical data, particularly for compounds like sulfide and methanol, are not available for our specific area or even nearby regions in the Tropical Eastern Pacific. This upwelling region is not well studied. We referenced the effects of upwelling in sediment sulfur cycling in other regions (Jørgensen et al. 2019; Kasten &

Jørgensen 2000; Dale et al. 2017) (L282-284), which are in line with our hypothesis of increased sulfide availability

In addition, in this revised version of the manuscript, we refer to further studies in the region that are consistent with our hypothesis of increased respiration and sulfide availability during the upwelling (Discussion section, L276-294):

- When seawater chemistry in Santa Elena Bay was compared between upwelling and non-upwelling seasons, it showed strong seasonal variation and a higher contribution of respiration to the CO₂ pool during upwelling (Noguera 2019).
- In the neighbouring Gulf of Papagayo, they found that the maximum chlorophyll concentration coincided with the minimum in oxygen concentrations of the water column (Cambronero-Solano et al. 2021)
- In nearby Golfo Dulce, increased abundances of sulfur-oxidizing bacteria and high sulfide oxidation rates during the dry season are consistent with the upwelling-induced increase in organic matter (Ferdelman et al. 2006)

On the other hand, methanol cycling in marine environments is poorly studied, particularly in the sediments, due to the low concentration and high solubility in the porewater (Fischer et al. 2021; Yanagawa et al. 2016). We could mainly find studies of the water column, and no methanol measurements in the TEP region. We have explicitly addressed this limitation in the discussion, particularly regarding our speculative interpretations of methanol dynamics during non-upwelling seasons (L303-305 and L316-322).

L229 The authors highlight that methanol and sulfide oxidation pathways are critical for symbiont metabolism under different environmental conditions, particularly during upwelling. Have these pathways also been observed in other ecosystems, such as cold seeps or hydrothermal vent regions?

We have revised the Introduction to provide a more comprehensive description of the symbiotic system and a detailed overview of the symbionts' metabolic potential (L75–95). Sulfide oxidation via the rDSR pathway is the canonical mechanism for energy acquisition in lucinid symbionts, and all symbionts described to date possess the potential for sulfide oxidation. In contrast, the capacity for methanol oxidation is relatively widespread in the symbiont genus *Ca. Thiodiazotropha*, but it is not universally present across all clades (Osvatic et al. 2021). Additionally, available expression data from lucinid symbionts in other environments, such as seagrass meadows and deep-sea ecosystems, show low to moderate expression of the methanol oxidation gene cluster (Lim et al. 2019; Ratinskaia et al. 2024; Osvatic et al. 2021). These findings suggest that methanol oxidation may play a lesser role in some environments compared to the TEP coast. We have further clarified this last point in the Discussion section (L326 - 329).

L257 As the authors note, additional physicochemical parameters are needed to strengthen the conclusions. Therefore, it is suggested to tone down the language in this subsection and in similar conclusions throughout the manuscript to reflect the current limitations.

We appreciate the Reviewer's suggestion to improve the precision of our manuscript. Given that we cannot conclusively establish that the observed transcriptional shifts are directly caused by variations in methanol and sulfide availability, we have toned down the language

throughout the manuscript to better reflect these limitations (e.g. revised title of Results section: Energy conservation pathways are differentially expressed ~~in response to upwelling~~ between upwelling and non-upwelling seasons).

We have also revised the title accordingly, which now reads: “Seasonal transcriptomic shifts reveal metabolic flexibility of chemosynthetic symbionts in an upwelling region.” As mentioned above, we have added a section to the Discussion that clearly addresses these limitations and highlights the need for future studies to integrate time-resolved geochemical and gene expression measurements (L316-322).

L174 & L233 There appears to be some overlap between the third and fourth sections. It seems this may result from combining results and discussion. To improve clarity, the authors might consider separating the results and discussion into distinct sections.

We agree with the reviewer’s recommendation to separate the Results and Discussion sections for greater clarity. Separating the Results and Discussion sections was an option we considered several times while writing the manuscript. Upon further reflection, we agree that this change improves the clarity of the manuscript, and so we have implemented it. We believe it has allowed us to elaborate on certain key points in the discussion while avoiding overlap. In addition, this has facilitated moving the section on core symbiotic genes from the Supplements to the Results section in the main text without disrupting the flow of the manuscript, which we believe allows for a more comprehensive interpretation of our results.

L288 The samples were transported at ambient temperature. How long was the time from sampling to freezing? Could this delay have affected the subsequent metagenomic and metatranscriptomic analyses?

The samples were immediately dissected and fixed in DNA/RNA Shield on site (Methods L350-354). DNA/RNA Shield, developed by Zymo Research, preserves nucleic acids at ambient temperature, eliminating the need for refrigeration or freezing during transport and storage (<https://www.zymoresearch.de/collections/dna-rna-shield?srsId=AfmBOorWCGu1tlu-LtlyGBVA2vif0qv7ZQCTg1g7DXX7G-Tzi5PwyY-d>). Although the samples were frozen after two weeks, this step was not necessary for preservation, and samples from both the upwelling and non-upwelling seasons were treated identically. Therefore, we do not believe that the transport or delay in freezing had any impact on the subsequent metagenomic and metatranscriptomic analyses.

L319 It would be helpful if the authors explicitly reported the final sequencing output for each metatranscriptome dataset.

This information is provided in S2 Table (metatranscriptomes, n_reads_raw). We also included a reference to this information in the Results section: “We analyzed the metatranscriptomes of ten lucinids sampled in the non-upwelling and upwelling seasons, respectively (metadata and statistics in S2 Table)” (L195-196).

L328 Typically, metagenomic sequencing outputs are higher than those for metatranscriptomes. However, in this study, the opposite trend was observed. Could the authors clarify the rationale behind this choice of sequencing output?

MAGs of lucinid symbionts in the area were available from previous studies, so obtaining new MAGs was not a priority. We estimated the necessary sequencing effort of the metagenomes to obtain reliable abundance and composition information from read mapping. We increased the sequencing effort of the metatranscriptomes because the focus of the study was transcriptional responses. Thus, we sequenced more deeply because typically metatranscriptomes contain a high fraction of host reads (a higher proportion than in the metagenomes).

The authors did not provide details about the biological replicates for the metatranscriptomes, which is a critical aspect.

We thank the reviewer for pointing out this oversight on our part. The number of replicates for metagenomic and metatranscriptomic libraries has now been explicitly stated in the methods section "libraries from each individual clam were constructed..." in L381 and L390-391. We also added this information to the Results section: "We analyzed the metatranscriptomes of ten lucinids sampled in the non-upwelling and upwelling seasons, respectively..." (L195-196).

Editorial review comments

This study investigates how chemosynthetic symbionts in *Ctena cf. galapagana* clams respond to seasonal environmental changes caused by coastal upwelling in Costa Rica. Using metagenomic and metatranscriptomic analyses, the authors compared symbiont communities and gene expression before and during the upwelling season. While the symbiont composition remained stable, both dominant symbiont clades shifted their gene expression in response to changing conditions. Methanol oxidation genes were more active before upwelling, while sulfur oxidation genes were upregulated during upwelling, reflecting changes in available energy sources. These findings highlight the metabolic flexibility of the symbionts and suggest this adaptability plays a key role in helping the host thrive in a dynamic environment.

We would like to express our sincere gratitude to the editor for taking the time to review this manuscript in the absence of a second reviewer. We believe that his suggestions have significantly improved the strength of our analysis and hypotheses. Please note that the line numbers correspond to the clean copy of the revised manuscript.

The introduction could be improved by giving a little more context into the functional capacity and lifestyles of the symbionts that were targeted via transcriptome analysis. This would give the readers a bit more background before getting into the specifics about "upwelling-induced" differential expression of the various pathways but with emphasis on the role of methanol and sulfide oxidation.

We appreciate the Editor's suggestion. We recognize that familiarity with our study system may have led to insufficient contextualization for readers less acquainted with these symbionts. Therefore, we have revised the Introduction to include a more comprehensive description of the symbiotic system and a detailed overview of the metabolic potential of the symbionts (L75–95). We hope these additions help readers better contextualize our findings.

The thing that seemed to be missing from the Introduction and early Results section were quantified measures of how the environment changed with respect to upwelling. There was very little info presented in the paper/figures with respect to environmental measurements, which could have included (for example) methanol/sulfide concentrations, temperature, pH, etc. The authors use the term "upwelling-induced" when referring to gene expression, however my view is that upwelling itself is not the driver of differential gene expression but rather the upwelling-associated changes in environment. This simple shift in organizing the Results and adding more context to the Introduction could add a lot to the interpretability of the Results.

Here are two suggestions of how the authors can better link observed changes in transcriptional behavior to environmental variability without having to run additional experiments:

i) Leverage existing environmental datasets more effectively or use other published geochemical profiles from the region to more explicitly correlate transcriptomic shifts with changes in substrate availability (e.g., sulfide or methanol levels) during upwelling vs. non-upwelling conditions.

As also addressed in the response to Reviewer 2's comment, we agree that having sediment geochemical profiles through time (in addition to satellite measurements of chlorophyll, nitrate and oxygen concentrations, and seawater temperature) would provide a more mechanistic understanding of the relationship between transcriptional shifts and dynamic fluctuations in substrate availability during upwelling and non-upwelling. However, to our knowledge, time series biogeochemical data, particularly for compounds like sulfide and methanol, are not available for our specific area or even nearby regions in the Tropical Eastern Pacific. This upwelling region is not well studied. We referenced the effects of upwelling in sediment sulfur cycling in other regions (Jørgensen et al. 2019; Kasten & Jørgensen 2000; Dale et al. 2017) (L282-284), which are in line with our hypothesis of increased sulfide availability

In addition, in this revised version of the manuscript, we refer to further studies in the region that are consistent with our hypothesis of increased respiration and sulfide availability during the upwelling (Discussion section, L276-294):

- When seawater chemistry in Santa Elena Bay was compared between upwelling and non-upwelling seasons, it showed strong seasonal variation and a higher contribution of respiration to the CO₂ pool during upwelling (Noguera 2019).
- In the neighbouring Gulf of Papagayo, they found that the maximum chlorophyll concentration coincided with the minimum in oxygen concentrations of the water column (Cambronero-Solano et al. 2021)

- In nearby Golfo Dulce, increased abundances of sulfur-oxidizing bacteria and high sulfide oxidation rates during the dry season are consistent with the upwelling-induced increase in organic matter (Ferdelman et al. 2006)

On the other hand, methanol cycling in marine environments is poorly studied, particularly in the sediments, due to the low concentration and high solubility in the porewater (Fischer et al. 2021; Yanagawa et al. 2016). We could mainly find studies of the water column, and no methanol measurements in the TEP region.

We toned down the language of our conclusions throughout the manuscript (e.g., revised title of Results section: Energy conservation pathways are differentially expressed ~~in response to upwelling~~ between upwelling and non-upwelling seasons) and added a section in the discussion that explicitly addressed these limitation in the discussion, particularly regarding our speculative interpretations of methanol dynamics during non-upwelling seasons, and highlights the need for future studies to integrate time-resolved geochemical and gene expression measurements (L303-305 and L316-322).

ii) Quantify and compare the expression levels of key metabolic genes relative to core housekeeping genes or total transcript counts to better estimate the relative contribution of different energy pathways under each condition, providing a clearer picture of symbiont metabolic adaptation as it relates to the interpretations around energy metabolism.

While normalizing by housekeeping genes is a common approach, we are concerned about introducing biases as housekeeping gene expression may not remain stable across environmental conditions due to changes in symbiont metabolic state, density, or host influence. Instead, we quantified the proportion of transcripts mapping to key metabolic genes involved in the reverse dissimilatory sulfite reduction (rDSR) pathway and the dissimilatory methanol oxidation pathway. Consistent with our hypothesis, genes associated with the rDSR pathway accounted for a higher percentage of total reads during upwelling conditions compared to non-upwelling conditions for both symbiont species. Specifically, for *Ca. T. boucheti*, rDSR genes represented $2.33 \pm 1.2\%$ of reads during upwelling versus $1.08 \pm 0.29\%$ during non-upwelling, a statistically significant difference (p -adjusted = 0.0047). For *Ca. T. endolucinida* TEP, the corresponding values were $2.41 \pm 0.75\%$ (upwelling) and $1.38 \pm 0.5\%$ (non-upwelling), though this trend was not statistically significant (p -adjusted = 0.27). In contrast, genes associated with methanol oxidation tended to be more abundant during non-upwelling conditions. In *Ca. T. boucheti*, these genes represented $5.59 \pm 3.12\%$ of reads during non-upwelling versus $1.12 \pm 0.32\%$ during upwelling (p -adjusted = 0.0103). A similar trend was observed in *Ca. T. endolucinida* TEP, with methanol oxidation genes comprising $6.03 \pm 3.78\%$ of reads during non-upwelling and $1.43 \pm 1.06\%$ during upwelling, although this difference was not statistically significant (p -adjusted = 0.0699). Although not all comparisons reached statistical significance, the observed patterns are consistent with shifts in potential metabolic activity across environmental conditions (see Figure below), with upwelling favoring sulfide oxidation and non-upwelling favoring methanol oxidation.

Despite these observed patterns, we ultimately chose not to include this analysis in the revised manuscript, as we believe it lacks the statistical rigor of the DESeq2-based differential expression approach. DESeq2 internally normalizes for sequencing depth and RNA composition, providing more robust comparisons of gene expression across conditions. Furthermore, while transcriptomic data provide useful insights into potential functional changes, they do not directly reflect actual pathway activity or energetic output. For this reason, we interpreted these results as indicative rather than definitive, and we present our conclusions as hypotheses (L313 and L303). We have also clarified this point in the revised Discussion, emphasizing the need for future work incorporating direct physiological or rate-based measurements (L318-322 and L337-339).

Other comments:

L180 "Since symbiont abundance and composition remained relatively constant between non-upwelling and upwelling samples (Fig. 2), the observed transcriptomic differences can be attributed to environmental changes, rather than factors like population size or interactions between the two symbiont clades." - since the authors have already performed PCA analysis, then it would be useful to perform a permanova analysis on this data to see if environmental change does indeed correlate more with the observed variance as compared to population size or any other measurements taken.

In response, we performed PERMANOVA to assess the contribution of season, symbiont abundance (TNSD), and the relative proportion of symbionts to the observed transcriptional variation across samples. Prior to conducting the PERMANOVA, we confirmed the assumption of homogeneity of dispersion (Methods, L472-475).

The PERMANOVA results indicate a statistically significant effect of the season in the transcriptional variation of both symbiont species. Additionally, we observed a statistically significant effect of symbiont relative abundance on transcriptional variation in *Ca. T. boucheti* (Results, L199-205). These findings support our conclusion that seasonal environmental changes contribute significantly to transcriptional dynamics, and they also allowed us to draw interesting perspectives on what else could be driving gene transcription in the symbionts (e.g. host control) (Discussion, L262-268). Nevertheless, our conclusions regarding the differences in the transcription of energy conservation pathways remain the same based on the differential expression analyses.

Minor comments:

There are several instances where the species names are not given in italics. E.g., L213

Thank you for pointing that out, however, we followed the International Code of Nomenclature of Prokaryotes when referring to *Candidatus* species, according to which only the term *Candidatus* (or *Ca.*) is italicized, while the genus and species names remain in regular font (Oren et al. 2023).

“A name of an organism in the status of *Candidatus* consists of the word *Candidatus*, followed by a name, based on one of the ranks defined in this Code (species, genus, family, etc.), formed in accordance with the nomenclature rules of the Code and its etymology appendix (...) Examples: *Candidatus* Methanoflorentaceae (family rank), *Candidatus* Methanoflorens (genus rank), *Candidatus* Methanoflorens stordalenmirensis (species rank). Note that the word *Candidatus*, but not the name that follows, is printed in italics.”

Furthermore, we have carefully revised all formatting inconsistencies throughout the manuscript.

REFERENCES

- Cambronero-Solano, S. et al., 2021. Hydrographic variability in the Gulf of Papagayo, Costa Rica during 2017-2019. *Revista de biología tropical*, 69(Suppl.2), pp.S74–S93. Available at: https://www.scielo.sa.cr/scielo.php?script=sci_arttext&pid=S0034-77442021000600074.
- Dale, A.W., Graco, M. & Wallmann, K., 2017. Strong and dynamic benthic-pelagic coupling and feedbacks in a coastal upwelling system (Peruvian shelf). *Frontiers in marine science*, 4. Available at: <http://journal.frontiersin.org/article/10.3389/fmars.2017.00029/full>.
- Ferdelman, T.G. et al., 2006. Biogeochemical controls on the oxygen, nitrogen and sulfur distributions in the water column of Golfo Dulce: an anoxic basin on the Pacific coast of Costa Rica revisited. *Revista de biología tropical*, 54(S1), pp.171–191. Available at: <https://revistas.ucr.ac.cr/index.php/rbt/article/view/26825>.
- Fischer, P.Q. et al., 2021. Anaerobic microbial methanol conversion in marine sediments. *Environmental microbiology*, 23(3), pp.1348–1362. Available at:

<http://dx.doi.org/10.1111/1462-2920.15434>.

Jørgensen, B.B., Findlay, A.J. & Pellerin, A., 2019. The Biogeochemical Sulfur Cycle of Marine Sediments. *Frontiers in microbiology*, 10, p.849. Available at: <http://dx.doi.org/10.3389/fmicb.2019.00849>.

Kasten, S. & Jørgensen, B.B., 2000. Sulfate Reduction in Marine Sediments. In H. D. Schulz & M. Zabel, eds. *Marine Geochemistry*. Berlin, Heidelberg: Springer Berlin Heidelberg, pp. 263–281. Available at: https://doi.org/10.1007/978-3-662-04242-7_8.

Lim, S.J. et al., 2019. Extensive Thioautotrophic Gill Endosymbiont Diversity within a Single *Ctena orbiculata* (Bivalvia: Lucinidae) Population and Implications for Defining Host-Symbiont Specificity and Species Recognition. *mSystems*, 4(4). Available at: <http://dx.doi.org/10.1128/mSystems.00280-19>.

Morel-Letelier, I. et al., 2024. Adaptations to nitrogen availability drive ecological divergence of chemosynthetic symbionts. *PLoS genetics*, 20(5), p.e1011295. Available at: <http://dx.doi.org/10.1371/journal.pgen.1011295>.

Noguera, C.S., 2019. *Carbonate chemistry and coral reefs in the Pacific coast of Costa Rica*. Doctoral Dissertation. Hamburg: Universität Hamburg. Available at: <https://ediss.sub.uni-hamburg.de/handle/ediss/6157>.

Oren, A. et al., 2023. International code of nomenclature of prokaryotes. Prokaryotic code (2022 revision). *International journal of systematic and evolutionary microbiology*, 73(5a). Available at: <https://doi.org/10.1099/ijsem.0.005585>.

Osvatic, J.T. et al., 2021. Global biogeography of chemosynthetic symbionts reveals both localized and globally distributed symbiont groups. *Proceedings of the National Academy of Sciences of the United States of America*, 118(29). Available at: <http://dx.doi.org/10.1073/pnas.2104378118>.

Ratinskaia, L. et al., 2024. Metabolically-versatile Ca. Thiodiazotropha symbionts of the deep-sea lucinid clam *Lucinoma kazani* have the genetic potential to fix nitrogen. *ISME communications*, 4(1), p.ycae076. Available at: <http://dx.doi.org/10.1093/ismeco/ycae076>.

Yanagawa, K. et al., 2016. Biogeochemical cycle of methanol in anoxic deep-sea sediments. *Microbes and environments*, 31(2), pp.190–193. Available at: <http://dx.doi.org/10.1264/jsme2.ME15204>.

Re: mSystems01686-24R1 (Seasonal transcriptomic shifts reveal metabolic flexibility of chemosynthetic symbionts in an upwelling region)

Dear Dr. Isidora Morel Letelier:

Your manuscript has been accepted, and I am forwarding it to the ASM production staff for publication. Your paper will first be checked to make sure all elements meet the technical requirements. ASM staff will contact you if anything needs to be revised before copyediting and production can begin. Otherwise, you will be notified when your proofs are ready to be viewed.

Sincerely,
Hans Bernstein
Editor
mSystems